# Clinical and Economic Impact in Dysphagia Management: A Preliminary Economic Evaluation for the WeanCare-Dysphameal Approach

**DOI:** 10.3390/nu17203259

**Published:** 2025-10-17

**Authors:** Chiara Monti, Paolo Landa, Antonio Rosario Romano, Marco Di Nitto, Axsinia Torsello, Stefania Ripamonti, Gianluca Catania, Annamaria Bagnasco, Milko Zanini

**Affiliations:** 1RSA Anni Azzurri Division KOS, 00137 Rome, Italy; 2Department of Operations and Decision Systems, Interuniversity Research Centre on Enterprise Networks, Logistics and Transport (CIRRELT), Université Laval, Québec, QC G1V 0A6, Canada; paolo.landa@fsa.ulaval.ca; 3Research and Development, Healthy Ageing Research Group, 25124 Brescia, Italy; 4Department of Health Sciences, University of Genova, 16132 Genoa, Italy; marco.dinitto@unige.it (M.D.N.); gianluca.catania@unige.it (G.C.); annamaria.bagnasco@unige.it (A.B.); 5Mirasole Anni Azzurri Residence KOS2, 00137 Rome, Italy; axsinia.torsello@anniazzurri.it; 6ASST della Brianza Dietetic Service P.O. of Desio U.O.S.D. Endocrine Diseases of Turnover and Nutrition, 20039 Desio, Italy; stefania.ripamonti@asst-brianza.it

**Keywords:** dysphagia, modified-texture diet, freeze-dried meals, cost-effectiveness, nutritional intervention, nursing homes, Dysphameal, WeanCare

## Abstract

**Background/Objectives**: Oropharyngeal dysphagia (OD) is a frequent condition among older adults in long-term care facilities, often leading to malnutrition, dehydration, and increased mortality. Modified-texture diets (MTDs) are used to address these risks, though traditional preparation methods often compromise nutritional density and standardization. The WeanCare protocol with Dysphameal^®^ technology introduces an innovative solution by offering IDDSI-compliant, high-density, freeze-dried meals. This study assesses its clinical effectiveness and economic impact. **Methods**: A six-month quasi-experimental pre–post study was conducted in two Italian nursing homes involving 13 older participants with diagnosed OD. Participants received all meals and hydration through Dysphameal^®^, and data were collected on anthropometry, biochemical markers, care workload, and economic parameters. Statistical analyses included regression, correlations, and pre–post comparisons. **Results**: Improvements were noted in fat-free mass (FFM), skeletal muscle mass (ASMM), and basal metabolic rate (BMR) in all participants. Albumin and lymphocyte counts improved, as did participant autonomy at mealtime. Operational time per participant dropped by 44% in kitchen and by 7 min/day in wards. Supplement use decreased, leading to significant cost savings and improved staff allocation. **Conclusions**: The WeanCare–Dysphameal^®^ system improves nutritional status, reduces caregiver workload, and enhances economic efficiency in institutional settings. It represents a scalable solution for OD management, ensuring consistency, safety, and sustainability in dysphagic care.

## 1. Introduction

Oropharyngeal dysphagia (OD) is a highly prevalent condition among older people, particularly those residing in long-term care facilities. Its prevalence ranges from 13% among functionally independent older adults to over 50% in institutionalized populations, as reported in recent national data from South Korea [1]. These findings are consistent with global estimates, which similarly describe OD as a widespread and underdiagnosed geriatric syndrome affecting up to 60% of institutionalized older individuals [2]. OD significantly increases the risk of malnutrition, dehydration, aspiration pneumonia, and mortality [3,4]. These complications not only compromise patient quality of life but also place a substantial burden on healthcare systems, both clinically and economically [5]. This underdiagnosed condition significantly worsens outcomes in frail older people, warranting structured and evidence-based interventions [1].

Orofaringeal dysphagia (OD) is a complex condition primarily resulting from neurodegenerative and structural impairments that compromise the coordination of swallowing [1,6]. In older or post-stroke patients, common mechanisms include neuromuscular dysfunction, reduced sensitivity, and sarcopenia, all contributing to delayed laryngeal closure and increased aspiration risk [1,6]. These physiological deficits are strongly associated with poor clinical outcomes, including recurrent respiratory infections and even mortality [7]. Early identification and tailored nutritional interventions are therefore critical to prevent complications and improve frail patient safety [6].

The pathophysiology of oropharyngeal dysphagia (OD) is multifactorial and primarily involves progressive neurodegenerative and structural changes that impair the complex coordination required for safe swallowing. Key mechanisms include lingual sarcopenia, reduced pharyngeal sensitivity, and neuromuscular dysfunction, which collectively delay laryngeal elevation and closure, impair hyoid bone excursion, and increase the risk of aspiration. These deficits are further exacerbated in frail or institutionalized older adults, where diminished muscle mass and reduced connective tissue elasticity compromise swallowing efficiency [1,6]. In stroke-related dysphagia, damage to cortical and subcortical swallowing centers disrupts the sensory–motor integration necessary for initiating and executing swallowing reflexes, with dysarthria and lower-limb motor deficits emerging as strong clinical indicators of impaired deglutition. Chronic microaspiration, a common consequence of dysphagia, is closely linked to pulmonary inflammation and recurrent respiratory infections, significantly elevating morbidity and mortality in this population [7]. As such, early recognition and structured nutritional interventions are essential to prevent deterioration in both physiological and functional domains, particularly as chronic aspiration contributes to systemic inflammation and infection in vulnerable populations [6].

To mitigate these risks, modified-texture diets (MTDs) have become a standard intervention for OD, aiming to reduce aspiration risk while maintaining adequate nutritional intake [4,8]. However, traditional methods—typically involving manual pureeing of cooked meals—are limited by low caloric and protein density, poor palatability, limited variety, and inconsistent texture, which together lead to reduced food consumption and compromised nutritional adequacy [5,9]. These shortcomings often necessitate the use of oral nutritional supplements (ONSs), thereby increasing healthcare costs and complicating care delivery [10]. In response to these clinical and operational challenges, the International Dysphagia Diet Standardisation Initiative (IDDSI) introduced a globally recognized framework that classifies foods and fluids into standardized texture and thickness levels, from Level 0 (thin liquids) to Level 7 (regular, easy-to-chew foods) [11]. While this system has improved safety and communication, its practical implementation remains uneven across settings due to the labor-intensive and subjective nature of manual texture modification [1,4]. Traditional blended meals frequently result in non-homogeneous textures characterized by inconsistent particle sizes and phase separation (e.g., solid–liquid stratification), which can be particularly dangerous for individuals with dysphagia. The presence of multiple consistencies within a single food item further elevates the risk of aspiration and diminishes the efficacy of the modified diet, ultimately compromising both nutritional intake and patient safety [4,9].

In recent years, technological innovations have advanced the field of dysphagia nutrition by enabling the development of industrially produced, standardized meals tailored to the specific needs of dysphagic patients. A notable example is the Dysphameal^®^ system, which employs freeze-dried or dehydrated food matrices reconstituted on demand using an automated dispensing unit. This process ensures precise texture conformity with the International Dysphagia Diet Standardisation Initiative (IDDSI) levels—primarily Levels 3 (liquidized) and 4 (pureed)—while maintaining consistent nutrient composition and long-term stability [12]. By removing variability from food preparation, the system aligns with food-first, patient-centered care models, supporting the dignity and autonomy of patients who rely on texture-modified diets [9]. Within institutional care contexts, the WeanCare protocol integrates this technology into a structured clinical and operational framework, facilitating its application in long-term and post-acute care facilities [13]. A mini-HTA (Health Technology Assessment) conducted by the Department of Health Sciences at the University of Genoa, with collaboration from multiple residential care institutions, identified several key advantages of Dysphameal^®^ over conventional modified-texture diets (MTDs). Among these are a significantly higher nutritional density—ranging from 1.2 to 1.3 kcal/g compared to 0.6 to 0.7 kcal/g in traditional blended meals [12]—and enhanced palatability and visual appeal, which improve voluntary oral intake and mealtime satisfaction [14]. The automation of meal preparation eliminates the need for manual blending, reducing the burden on kitchen and clinical staff and minimizing risks associated with texture inconsistency such as grittiness or phase separation [12]. Moreover, the meals’ ability to be stored at room temperature and dispensed in modular portions optimizes logistical management and supports rapid, hygienic service delivery in institutional settings [12]. Importantly, the HTA findings also highlight how Dysphameal^®^ can reduce dependence on oral nutritional supplements (ONSs), mitigate nutrition-related complications such as dehydration and constipation, and enhance patient satisfaction and nutritional autonomy. These outcomes are further substantiated by recent real-world observational data from facilities employing the WeanCare–Dysphameal^®^ system, where participants demonstrated measurable improvements in muscle mass, hydration status, resting metabolic rate, and immunological biomarkers, alongside substantial time and cost savings in both kitchen operations and ward-level care delivery [13,15]. Despite the growing body of evidence supporting texture-modified nutritional interventions, few studies have simultaneously addressed the clinical efficacy and the economic–organizational impact of such technologies. The present study aims to fill this gap by evaluating the WeanCare–Dysphameal^®^ system across two Italian residential care facilities, focusing on improvements in nutritional status, care delivery efficiency, and cost-effectiveness.

## 2. Materials and Methods

### 2.1. Study Design

This was a quasi-experimental, pre–post observational study conducted over six months in two Italian residential care facilities (Residenze Sanitarie Assistenziali, RSA).

This observational study was conducted and reported in accordance with the Strengthening the Reporting of Observational Studies in Epidemiology (STROBE) guidelines [16].

The study was approved by the Ligurian Territorial Ethical Committee (CET Liguria), protocol number 16/2020 (Study number 677/2020), on 16 December 2020, as part of the broader WeanCare project.

The primary objective was to evaluate the organizational impact of a standardized, texture-modified nutritional protocol based on the Dysphameal^®^ system, applied to institutionalized participants with oropharyngeal dysphagia (OD). The study was conducted in real-world conditions to reflect typical operational and participant care dynamics. Participants were involved due to clinical and nutritional confirmation, as in previous research, of the effectiveness of the protocol on their side, and no control group was included because each participant served as their own baseline comparator.

### 2.2. Study Population

A total of 13 elder residents (≥75 years) were enrolled based on the following inclusion criteria:Clinical diagnosis of oropharyngeal dysphagia, categorized as IDDSI level 3 (Liquidized) or level 4 (Pureed);Documented risk or evidence of malnutrition (via weight loss, reduced food intake, or clinical indicators);Continuous residence in the facility for at least three months prior to enrollment;Provision of informed consent by the participant or their representative.

Exclusion criteria included:End-of-life status or palliative care indication;Need for enteral nutrition (PEG or nasogastric tube);Severe acute infections at baseline.

### 2.3. Intervention and Data Collection Procedures

The intervention has been promoted by the institution involved. The study had to observe output and outcome of the modification in food preparation and administration. Activities reported were based on the exclusive administration of meals and hydration products prepared using the Dysphameal^®^ system in some of the guests of the residential facilities.

Each participant received three main Dysphameal-modified meals per day, corresponding to breakfast, lunch, and dinner, and a snack in the middle of the afternoon, in accordance with their individual energy requirements assessed by the facility’s dietitian. The average caloric provision was approximately 1800 kcal/day, with macronutrient distribution adapted to meet IDDSI texture and consistency standards. Hydration was supported with pre-thickened gelled water provided ad libitum, and no oral nutritional supplements were administered unless previously prescribed.

This technology involves dehydrated or freeze-dried food matrices that are reconstituted and emulsified via a semi-automated dispenser, precisely calibrated to deliver consistencies aligned with the International Dysphagia Diet Standardisation Initiative (IDDSI), specifically Levels 3 and 4. The reconstruction and emulsification process was performed collectively once for all participants for each main meal. Participants received all main meals (breakfast, lunch, dinner) as well as snacks and hydration products, including Level 3 IDDSI-compliant gelled water or thickeners. The system allows controlled incorporation of water and vegetable oil to meet defined rheological and caloric specifications, ensuring a high nutritional density (approximately 1.2–1.3 kcal/g), texture stability without phase separation, and minimal infrastructure requirements due to its modular and hygienic design. Prior to the intervention, multidisciplinary training sessions were conducted for all involved personnel—including nurses, dietitians, speech–language pathologists, and food service staff—to ensure consistent protocol adherence and compliance with IDDSI guidelines.

Data were collected at two time points: baseline (T0) and after six months (T6). Measurements were grouped into four domains. First, anthropometric and body composition parameters were assessed using bioelectrical impedance vector analysis (BIVA), capturing data on body weight, fat-free mass (FFM), skeletal muscle mass (ASMM), fat mass, fat-free mass index (FFMI), total body water (TBW), and basal metabolic rate (BMR). Second, biochemical parameters were obtained through routine blood tests, including albumin, total proteins, lymphocyte count, cholesterol, transferrin, C-reactive protein (CRP), and micronutrients such as vitamin D, vitamin B12, folate, iron, and creatinine. Third, functional and clinical indicators were evaluated using tools such as the EdFed Scale for feeding difficulties in dementia, the number of enemas administered per participant, level of autonomy during meals (e.g., spoon-feeding needs), and food intake assessed by quartiles of portion consumption. Lastly, organizational and economic parameters included kitchen- and ward-level meal preparation time per participant, use of oral nutritional supplements (ONSs), staff workload per shift, and monthly participant-level cost metrics related to supplementation, labor, and food waste.

The selected variables were chosen to reflect the multidimensional nature of nutritional and functional health in older adults with dysphagia. Anthropometric and body composition parameters (e.g., ASMM, FFM, BMR, TBW) were prioritized due to their known association with sarcopenia, hydration status, and metabolic recovery. Biochemical markers, although influenced by inflammation, were included to provide additional insight into systemic nutritional status. Functional outcomes, such as enema use and feeding autonomy, were selected as practical indicators of digestive function and care dependency. These domains align with established geriatric nutrition frameworks and were chosen to capture the clinical, operational, and economic effects of the intervention, in accordance with ESPEN recommendations for multidimensional assessment in older adults [17].

No structured physical activity or exercise therapy programs were introduced during the intervention period. Participants continued with their usual level of physical and recreational activities, which were comparable across the cohort, including light physiotherapy, routine mobilization, and social engagement activities typical of the facility’s standard care.

### 2.4. Statistical Analysis

Descriptive statistics were reported as means and standard deviations. Changes from baseline to follow-up were evaluated using paired *t*-tests (α = 0.05).

Pearson correlation analysis was used to explore associations between:ASMM and BMR;TBW and ASMM;FFMI and weight.

Multiple linear regression models were constructed to identify the strongest predictors of improvement in skeletal muscle mass (ASMM), with explanatory variables including FFM, FM, BMR, FFMI, and TBW.

Regression model diagnostics included R^2^ and *p*-values. Significance was defined as *p* < 0.05, and all analyses were performed using standard statistical packages (e.g., SPSS v.28-2021 or R v. r4.1.0 2021).

## 3. Results

The results are organized into three domains: clinical–nutritional outcomes, functional and autonomy-related measures, and organizational–economic outcomes. Results are presented across three domains—clinical–nutritional, functional–autonomy, and organizational–economic outcomes—to reflect the multidimensional goals of the intervention. This structure mirrors the study design and allows a comprehensive interpretation of the effects across physiological, behavioral, and operational parameters. All 13 participants completed the 6-month follow-up, and no adverse events were reported.

### 3.1. Clinical and Nutritional Improvements

#### 3.1.1. Anthropometric and Body Composition Outcomes

All participants exhibited improvements in key indicators of nutritional status.

Fat-Free Mass (FFM) increased in 100% of participants (mean FFM = 14.7 kg, SD = 1.85 kg). Skeletal Muscle Mass (ASMM) improved in all participants (mean ASMM = 6.5 kg, SD = 1.2 kg). Fat-Free Mass Index (FFMI) and Basal Metabolic Rate (BMR) showed positive trends in 13/13 cases. Significant correlations were observed: ASMM strongly correlated with BMR (r = 0.94) and TBW (r = 0.98); FFM and ASMM correlated strongly with final weight (r = 0.83); Linear regression between TBW and ASMM showed an adjusted R^2^ = 0.96 (*p* < 0.001).

#### 3.1.2. Biochemical Markers

Biochemical improvements were documented (Table 1):Albumin increased in 63.6% of participants (mean from 3.14 g/dL to 3.42 g/dL);Lymphocyte count increased in 40% of participants, indicating enhanced immunological function;Total cholesterol improved, lowering in 70% of participants;Mild improvements were noted in transferrin and CRP levels.

These trends suggest reduced inflammatory status and improved nutritional homeostasis, supporting previous findings on the impact of texture-modified, high-density diets.

Although albumin and transferrin are recognized as nonspecific biomarkers influenced by systemic inflammation, their role in this observational study was carefully framed. Blood samples were collected during clinically stable periods, ensuring that no acute infections or inflammatory events were recorded in patient charts, and CRP values remained within normal reference ranges. In long-term residential care settings, especially among frail older adults, visceral proteins—though imperfect—can yield contextual insights when evaluated alongside anthropometric, functional, and dietary measures.

According to the ASPEN position paper, albumin and prealbumin should not be used as sole proxies for nutritional status because their levels decline in the presence of inflammation irrespective of true protein-energy malnutrition [18]. Nonetheless, emerging evidence from large cohorts indicates that dynamic changes in albumin—especially increases over time in stable inflammatory conditions—may reflect improvements in the interplay between nutrition and inflammation [19]. Therefore, in our study, albumin and transferrin are interpreted not as isolated nutritional biomarkers but as elements within a multidimensional nutritional assessment, consistent with real-world practice in long-term care.

### 3.2. Functional and Autonomy-Related Outcomes

The intervention also impacted functional indicators of nutritional well-being:The average number of enemas per participant/month decreased from 1.25 to 0.43 (−66%) (Table 2);Feeding autonomy improved: 13 participants no longer required assisted spoon-feeding;EdFed scores declined across the cohort, reflecting reduced feeding difficulties;Average meal intake increased (assessed via quartile analysis), with over 80% of daily portions consumed spontaneously.

### 3.3. Functional and Autonomy-Related Outcomes

#### 3.3.1. Time Savings in Kitchen and Ward

In the kitchen, the average meal preparation time per participant decreased from 6.24 to 3.58 min—a reduction of 44.83%. On the wards, approximately 7 min per participant per day was saved during breakfast and hydration routines, primarily due to replacing manual thickener mixing with pre-set gelled water. The total daily time savings are detailed in Table 3.

#### 3.3.2. Supplement and Cost Reductions

All participants discontinued oral nutritional supplements, resulting in an average reduction of 386 kcal and 21.3 g of protein per participant per day. We used the following estimated labor costs: €0.2955 per minute for a nurse (from the regulatory board for public work contracts ARAN, 2022 [20]), €0.2695 per minute for a kitchen chef, and €0.2270 per minute for a sous-chef (FIPE, 2024 [21]). For fresh food preparation, we assumed that 33% of the workload is handled by the kitchen chef and 67% by the sous-chef. In contrast, for the Dysphameal^®^ system, we assumed 20% of the workload is managed by the kitchen chef and 80% by the sous-chef. A comparative table of the costs and preparation times for both fresh food and Dysphameal^®^ is provided in Table 3. All cost estimates are expressed in 2025 euros, adjusted for inflation based on the appropriate price indices for values originally calculated before 2025 (ISTAT, 2025 [22]).

We assumed that the cost of fresh food and Dysphameal^®^ meals is equivalent, based on meal cost estimates from a previous study (Sebastiano, 2017 [23]). While the actual price of Dysphameal^®^ is not currently available, it is expected to be similar to the manufacturer’s quoted price. Due to this uncertainty, we conducted a sensitivity analysis comparing the prices of fresh food and Dysphameal^®^ to better understand the potential cost savings of the intervention relative to the comparator. Electricity and equipment usage costs were derived from WeanCare study. Food thickener is required only for fresh food to ensure appropriate consistency and texture for participants. We assumed enema usage to be the same for both Dysphameal^®^ and fresh food, with a daily cost of €0.05 per participant (Weancare).

The results show a total daily cost of €19.05713 (or €6955.85 annually) for the fresh food and €13.93640 per day (or €5086.79 annually) for Dysphameal^®^. This represents a daily saving of €5.12073, amounting to €1869.06 annually with the use of Dysphameal^®^.

From an organizational standpoint, Table 3 and Table 4 quantify the time and cost savings derived from streamlining meal preparation and eliminating oral nutritional supplements. These metrics reveal not only operational efficiency but also opportunities for reallocating clinical resources toward more complex care needs. Taken together, the integrated clinical, functional, and economic data affirm the Dysphameal^®^ system as a viable and scalable innovation in dysphagia care.

#### 3.3.3. Sensitivity Analysis on Disphameal^®^ and Fresh Food Costs

As mentioned above, the absence of precise pricing information for Dysphameal^®^ introduces uncertainty into the estimation of potential cost savings from adopting the intervention. To address this, a sensitivity analysis was conducted to evaluate how variations in the daily cost difference between Dysphameal^®^ and fresh food affect overall savings for feeding a single participant.

The Table 5 shows that Dysphameal^®^ remains a cost-saving option as long as its price does not exceed that of fresh food by more than €5.12 per participant per day. At a cost difference of exactly €5.12, both options result in equivalent total costs. If the cost of Dysphameal^®^ exceeds fresh food by more than €5.12 per day, then fresh food becomes the more economical choice. This sensitivity analysis helps quantify the financial threshold at which Dysphameal^®^ transitions from a cost-saving to a cost-incurring intervention.

## 4. Discussion

The findings from this study confirm the significant clinical, functional, and organizational benefits of implementing the WeanCare protocol using Dysphameal^®^ technology in long-term care settings for older patients with oropharyngeal dysphagia (OD). By aligning with the IDDSI framework and leveraging a standardized, high-nutrient meal delivery system, the intervention addressed key limitations of traditional modified-texture diets (MTDs), including poor consistency, inadequate nutrient density, and labor-intensive preparation.

### 4.1. Nutritional and Clinical Improvements

As anticipated, the nutritional outcomes reflect the central tenets of the intervention’s rationale: improved intake leads to improved body composition and metabolic recovery. Consistent with the pathophysiological understanding that OD patients are at high risk of malnutrition and sarcopenia [1,3,4], we observed significant improvements in fat-free mass (FFM), skeletal muscle mass (ASMM), and basal metabolic rate (BMR). The strong correlations between ASMM and both total body water (TBW) and BMR support existing evidence that muscular recovery is linked not only to macronutrient intake but also to adequate hydration status and metabolic stimulation [6,8,9]. These anthropometric changes are particularly relevant in institutionalized older adults with dysphagia, who are at high risk of sarcopenia and undernutrition.

While increases in biochemical markers such as albumin and lymphocyte counts were noted in a proportion of participants, these findings must be interpreted with caution. Albumin and transferrin are known to be influenced by inflammatory status and hydration, and are therefore not definitive indicators of nutritional status when considered in isolation. In this study, their inclusion aimed to provide contextual biochemical data, which were interpreted alongside CRP levels and clinical stability to minimize confounding.

Importantly, these improvements were achieved without the use of external oral nutritional supplements (ONSs), which were completely discontinued. This is a clinically relevant finding: not only does it reflect the efficacy of the Dysphameal^®^ meals in delivering caloric and protein needs (up to 1.3 kcal/g), but it also aligns with recent literature advocating for integrated, food-first approaches over supplemental nutrition in institutionalized older adults [10,11]. This supports growing evidence that integrated, non-supplement-based interventions can effectively meet protein-energy needs [9].

Biochemical markers corroborated the anthropometric improvements. Albumin levels increased in nearly two-thirds of participants, while lymphocyte counts and cholesterol values improved in substantial proportions. Though not all biochemical parameters improved universally—iron and folate levels remained low in several participants—these findings may reflect underlying comorbidities, absorption issues, or long-standing deficiencies that require targeted interventions beyond nutritional texture standardization alone. These gains mirror outcomes from recent studies demonstrating improved muscle strength and metabolism in dysphagic participants receiving nutrient-dense, care-optimized meals [14].

### 4.2. Functional Recovery and Participant Autonomy

Feeding autonomy is a critical quality-of-life indicator in dysphagic populations, particularly among individuals with cognitive impairment or physical frailty. The WeanCare–Dysphameal^®^ protocol notably reduced the number of participants requiring spoon-feeding, an outcome with both clinical and ethical implications. This change, supported by improved EdFed scores, reflects the role of sensory recognition, palatability, and standard texture in enhancing spontaneous food intake and promoting independence [12].

Moreover, the significant reduction in enema use suggests improved gastrointestinal motility, likely attributable to better hydration and fiber intake inherent in the standardized, rehydrated meals and gelled water systems. This outcome is not trivial: constipation is a common and often overlooked burden in older adults, frequently leading to discomfort, polypharmacy, and nursing intervention time [14].

### 4.3. Organizational and Economic Impact

One of the most compelling findings of this study lies in its implications for operational efficiency. Preparation time for dysphagic meals in the kitchen decreased by 44%, with a further 7 min/day/participant saved in the ward, primarily in hydration and breakfast routines. In facilities with limited staffing and increasing demands, such time savings—amounting to nearly 3 h/day for 20 dysphagic participants—translate into tangible organizational benefits. Time reduction in foodservice tasks aligns with reports from other care homes adopting semi-automated dysphagia protocols [12,13].

While it is true that part of the observed time savings—specifically the 7 min/day/participant during hydration and breakfast routines—are due to the use of pre-set gelled water instead of manual thickener mixing, this component alone does not fully explain the total operational gains. The Dysphameal^®^ system integrates gelled hydration as part of a comprehensive and standardized delivery protocol, ensuring consistent rheological properties, portioning, and readiness alongside meal components. In contrast, adopting pre-set gelled water in isolation within conventional foodservice systems would require separate procurement, storage, and staff training protocols, and may lack integration with the broader IDDSI consistency framework applied in Dysphameal^®^.

Furthermore, within our implementation, the preparation of meals and hydration products was centralized and semi-automated, enabling parallel time savings not easily replicated in traditional kitchen workflows. Therefore, while gelled water alone could contribute to partial time savings, the cumulative impact observed in this study reflects the synergistic effect of the integrated Dysphameal^®^ protocol, rather than any single component.

Furthermore, the ability to delegate preparation tasks to non-specialized staff—after minimal training—represents a shift in workforce strategy, especially valuable in contexts where experienced culinary or clinical personnel are scarce during certain shifts. This aspect enhances the system’s scalability across various residential care settings.

These results collectively reinforce the view that Dysphameal^®^ contributes to both clinical and economic efficiencies. Specifically, the observed improvements in nutritional biomarkers—such as increased albumin and lymphocyte counts—suggest a systemic enhancement of nutritional and immune status, likely driven by the higher protein and caloric density of the meals.

The relationships visualized in Figure 1 and Figure 2 (e.g., between skeletal muscle mass and hydration status, and between muscle mass and basal metabolic rate) highlight the protocol’s physiological coherence and underline the role of optimized hydration and nutrient delivery in supporting musculoskeletal recovery in frail older adults.

Beyond physiological effects, the intervention produced measurable benefits in daily care operations. As shown in Table 2, the frequency of enemas decreased markedly—likely reflecting improved gastrointestinal function and fiber intake. Our observation suggests that better hydration and body composition can reduce reliance on symptomatic interventions.

Additionally, the increase in feeding autonomy—evidenced by fewer participants requiring spoon-feeding—demonstrates the potential of Dysphameal^®^ to enhance dignity, reduce caregiver workload, and streamline ward-level care.

Lastly, the sensitivity analysis conducted on the comparative cost of fresh food and Dysphameal^®^ quantifies the economic threshold at which the latter ceases to offer financial savings, thereby offering a pragmatic tool for decision-makers in evaluating the intervention’s sustainability.

From an economic standpoint, the intervention eliminated the need for ONSs and reduced nursing time dedicated to feeding and symptom management, reinforcing that the clinical gains of Dysphameal^®^ are effective, considering health outcomes and cost-saving as it reduces the overall costs with respect to the manual preparation of the meals using fresh ingredients.

### 4.4. Integration with Existing Literature

These findings build upon previous studies highlighting the limitations of traditional blended diets and the value of standardized texture-modified meals [13,15,24]. Ballesteros-Pomar et al. [4] emphasized that conventional MTDs often fail to meet both safety and nutritional adequacy criteria, especially when prepared manually. Our results directly confirm and expand upon these concerns by demonstrating measurable improvements across nutritional, functional, and organizational domains when standardized, IDDSI-compliant meals are implemented.

Moreover, the observed improvements in ASMM and hydration (TBW) parallel the findings of Vucea et al. [10], who associated better nutritional profiles with decreased reliance on symptomatic interventions like enemas and ONSs. Our study adds a pragmatic dimension to these associations by quantifying operational and cost reductions in real-world settings.

Our findings corroborate those of prior WeanCare implementations, which also reported improved caregiver efficiency and nutritional outcomes [15,24].

### 4.5. Limitations and Future Directions

While the present findings are operationally significant and support the clinical utility of the WeanCare–Dysphameal^®^ system, certain limitations should be acknowledged. The relatively small sample size and the lack of a control group may constrain statistical power and limit the generalizability of results, particularly regarding clinical endpoints. However, it is important to contextualize this study within its primary aim: to evaluate the organizational and economic implications of transitioning from traditional blended texture-modified foods (TMFs) to a standardized, industrially prepared alternative. The focus of this investigation was not to re-establish the clinical efficacy of the Dysphameal^®^ protocol—which has been rigorously documented in earlier research, including our previously published study—but rather to monitor implementation fidelity and confirm that clinical and nutritional outcomes remain consistent with those prior findings. The current cohort was followed to verify alignment with those established quality of life and health-related benefits. While the use of regression and correlation analyses in small samples may raise concerns about statistical power and generalizability, it is important to clarify that the primary aim of this study was not to establish causality or make population-level inferences. Rather, the analyses were conducted in an exploratory manner to observe potential relationships between nutritional, metabolic, and functional parameters within a defined, real-world cohort.

All eligible residents meeting the inclusion criteria were enrolled, and no additional participant selection was applied. This approach aligns with the pragmatic nature of the study, which focused primarily on the comparative operational and economic impact of Dysphameal^®^ versus traditional blended diets. Therefore, the statistical findings should be interpreted cautiously and viewed as hypothesis-generating rather than definitive. Nonetheless, additional large-scale, multi-center studies—preferably randomized controlled trials—are warranted to explore long-term sustainability, participant-reported outcomes (e.g., satisfaction, sensory acceptance), and broader applicability across diverse institutional settings.

### 4.6. Practical Implications

The findings from this study suggest several practical applications for long-term care settings managing residents with dysphagia. First, the integration of ready-to-use, texture-modified meals such as Dysphameal^®^ can reduce the need for individualized meal preparation, ensuring consistency in IDDSI levels while lowering kitchen workload and human error risk.

Second, the inclusion of gelled hydration products as part of a standardized feeding protocol streamlines care routines, shortens administration times, and minimizes variability in intake—a frequent issue in dysphagic patients.

Third, the observed improvements in functional outcomes, such as feeding autonomy and bowel regularity, indicate that tailored nutritional strategies can positively impact patient independence and reduce caregiver burden.

These outcomes are particularly relevant in contexts with limited staff-to-patient ratios and high prevalence of multimorbidity. Future implementations could benefit from integrating such interventions into routine nutritional care, supported by multidisciplinary teams to ensure sustainability and monitoring.

## 5. Conclusions

The implementation of the WeanCare protocol using Dysphameal^®^ technology provides a structured and potentially effective approach to managing oropharyngeal dysphagia in institutionalized older adults. In this pilot study, the intervention was associated with improvements in participants’ nutritional intake, body composition, and functional autonomy, while minimizing the use of oral nutritional supplements—thanks to improved intake from standardized, nutrient-dense meals—and reducing the incidence of constipation.

Beyond these clinical observations, the protocol also showed operational benefits, including time savings in kitchen and ward routines and more flexible staff workflows, indicating potential economic advantages in long-term care settings.

By delivering standardized, IDDSI-compliant meals with high nutritional density, the WeanCare–Dysphameal^®^ system may help address some limitations of traditional modified-texture diets and offer a scalable, safe, and economically sustainable option for dysphagia management.

However, given the small sample size and exploratory nature of this study, these findings should be interpreted with caution and considered hypothesis-generating. Further research with larger cohorts and inclusion of quality-of-life, cost-effectiveness, and satisfaction outcomes is needed to confirm and extend these preliminary results.

## Figures and Tables

**Figure 1 nutrients-17-03259-f001:**
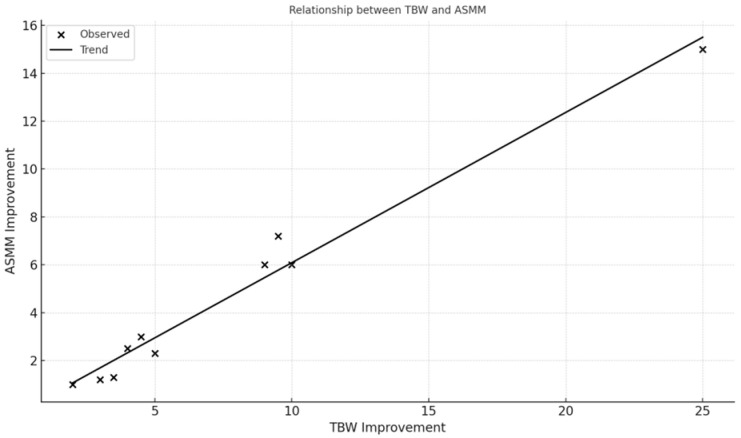
Linear relationship between total body water (TBW) improvement and appendicular skeletal muscle mass (ASMM) gain over the intervention period; Observed values (x) and trend line indicate a positive association between hydration status and muscle mass gain following the Dysphameal intervention.

**Figure 2 nutrients-17-03259-f002:**
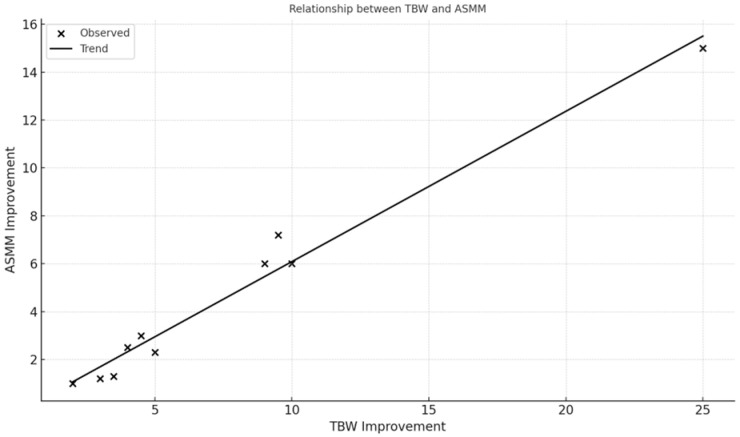
Positive correlation between appendicular skeletal muscle mass (ASMM) improvement and basal metabolic rate (BMR) increase. Observed data points (x) and fitted linear trend line indicate a proportional relationship between muscle mass gain and resting energy expenditure following the Dysphameal dietary intervention.

**Table 1 nutrients-17-03259-t001:** Biochemical markers.

Parameter	Participants Improved (n, %)	Participants Worsened (n, %)
Albumin	7 (63.64%)	4 (36.36%)
Total Proteins	3 (75.00%)	1 (25.00%)
Transferrin	5 (50.00%)	5 (50.00%)
Creatinine	3 (100.00%)	0 (0.00%)
Iron	3 (30.00%)	7 (70.00%)
C-Reactive Protein (CRP)	1 (33.33%)	2 (66.67%)
Lymphocytes	4 (40.00%)	6 (60.00%)
Cholesterol	8 (72.73%)	3 (27.27%)
Vitamin B12	3 (33.33%)	6 (66.67%)
Folate	2 (22.22%)	7 (77.78%)
Vitamin D	5 (55.56%)	4 (44.44%)

**Table 2 nutrients-17-03259-t002:** Number of Enemas.

Number of Enemas	Mean Value	Standard Deviation
Baseline Mean Frequency	1.25	1.16
Final Mean Frequency	0.43	0.53
Mean Reduction	0.86	1.35

**Table 3 nutrients-17-03259-t003:** Time savings Kitchen/Ward.

Area	Time/Participant/Day (Pre)	Time/Participant/Day (Post)	Time Saved	% Reduction
Kitchen	6.49 min	3.58 min	2.91 min	−44.84%
Ward (hydration and breakfast)	18.5 min	11.5 min	7.0 min	−37.84%
Total per day	24.99 min	15.08 min	9.91 min	−39.66%
Total per year	912,135 min	550,420 min	361,715 min	

**Table 4 nutrients-17-03259-t004:** Comparative table for Fresh food and Disphameal^®^.

Description	Fresh Food	Disphameal^®^	Unit Measure	Source
Preparation Kitchen	6.49	3.58	min	WeanCare
Ward routine	18.5	11.5	min	WeanCare
Food supplements	1.11	0.05	Euro	WeanCare
Food cost	9.564828	9.564828	Euro	Assumption
Electric consumption	0.175	0.03	Euro	WeanCare
Machinery	0.125	0	Euro	WeanCare
Food thickener	1	0	Euro	WeanCare
Enema	0.05	0.05	Euro	WeanCare

**Table 5 nutrients-17-03259-t005:** Sensitivity analysis on daily cost difference between Dysphameal^®^ and fresh food.

Cost Difference Between Disphameal^®^ and Fresh Food	Daily Cost Savings (Euros)	Yearly Cost Savings (Euros)	Cost Saving Product
−5 euros	10.12072503	3694.064635	Disphameal^®^
−4 euros	9.120725027	3329.064635	Disphameal^®^
−3 euros	8.120725027	2964.064635	Disphameal^®^
−2 euros	7.120725027	2599.064635	Disphameal^®^
−1 euro	6.120725027	2234.064635	Disphameal^®^
0 euro	5.120725027	1869.064635	Disphameal^®^
1 euro	4.120725027	1504.064635	Disphameal^®^
2 euros	3.120725027	1139.064635	Disphameal^®^
3 euros	2.120725027	774.0646347	Disphameal^®^
4 euros	1.120725027	409.0646347	Disphameal^®^
5 euros	0.120725027	44.06463473	Disphameal^®^
6 euros	−0.879274973	−320.9353653	Fresh food
7 euros	−1.879274973	−685.9353653	Fresh food

## Data Availability

The data supporting the findings of this study are not publicly available due to commercial confidentiality agreements. However, the data may be made available by the authors upon reasonable request, subject to the approval of the entities holding the proprietary rights.

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
