# Peer review of "Clinical and Economic Impact in Dysphagia Management: A Preliminary Economic Evaluation for the WeanCare-Dysphameal Approach"

_nutrients, 2025, doi:10.3390/nu17203259_

Round 1

Reviewer 1 Report

Comments and Suggestions for Authors

The authors performed a study with the aim to examine the clinical and economical impact of a pre-packed freeze-dried IDDSI specific meal for adults with dysphagia living in nursing homes. They followed 13 adult patients for 6 months, collected data regarding body composition and biochemical markers as surrogate markers of nutritional status, feeding independence and enema requirements as surrogate markers for functional indicators and finally a cost analysis was performed to determine whether this intervention is cost effective. The results of the study are encouraging. However, given the small sample size, negating statistical analysis in most comparisons, the conclusions that can be drawn from this study are limited. while the authors acknowledge this in the limitation section, this limitations needs to be reflected in the body of the discussion, with "toning-down" the conclusions so they will fit the scope of the results. 

In addition, there is a typing error in line 156 - istituion instead of institution.  

Author Response

We sincerely thank the reviewer for the thorough evaluation of our manuscript and for the constructive suggestions provided. Your insightful comments have been invaluable in helping us to improve the clarity, methodological transparency, and overall quality of the paper. We greatly appreciate the time and expertise you dedicated to this review, and we have carefully revised the manuscript in light of your recommendations.

Reviewer 1

Conclusions should be toned down in the discussion due to small sample size.

Thank you for this helpful observation. The Discussion section (Page 15, Line 2) was modified to emphasize the exploratory nature of the findings. Language has been softened accordingly to better reflect the limitations inherent to the small sample size.

Page 15, Line 2

Reviewer 1

Typo: 'istitution' should be 'institution'.

Thank you for catching this error. The typo has been corrected from 'istitution' to 'institution'.

Page 8, Line 1

Reviewer 2 Report

Comments and Suggestions for Authors

To the Authors,

Thank you for opportunity to review this article for publication in Nutrients.

This study examines the clinical and economic impact of a specialized meal for older adults with dysphagia, which is an important and relevant topic.

The following concerns should be addressed before the manuscript can be considered for publication:

Overall Comments:

Since the participants were residents of long-term care facilities, the term participant is more appropriate than patient.

Introduction:

# The background and previous reports are well summarized. However, as this study aims to evaluate the effects of a specific dysphagia-modified meal, the detailed description of swallowing mechanisms in the first half of the introduction appears unnecessarily lengthy.

Methods:

# As this was an observational study, please state explicitly whether it was conducted in accordance with the STROBE statement.

# Please explain how the 6-month intervention period was determined and whether this duration is reasonable.

# Information regarding the amount of food provided during the intervention is missing. Since nutritional status is one of the outcomes, the criteria for food provision should be clearly described.

# Regarding Dysphameal preparation: was the reconstruction and emulsification performed simultaneously for all 13 participants or individually for each person? This has implications for workload and cost; please provide details.

# Skeletal muscle mass was included as an outcome, but were exercise therapy or recreational activities provided at the same frequency for all participants?

Results:

# Please add titles to all figures.

# The axes of the figures are unclear. Please specify what is represented, include appropriate units, and add the figure legends accordingly.

# For Table 1 and Table 2:

Titles should be placed above the tables.

Tables should be formatted to fit within a single page.

# Albumin and transferrin are not recommended as nutritional markers since they are influenced by inflammation. Please reconsider whether these parameters are appropriate as outcomes.

# The manuscript states: “On the wards, approximately 7 minutes per patient per day were saved during breakfast and hydration routines, primarily due to replacing manual thickener mixing with pre-set gelled water.” Is this a feature specific to Dysphameal? Would similar time savings not also be achieved if gelled water were routinely used in conventional meals?

# Sections 3.3.2 and 3.3.3 include interpretations of results. These should be moved to the Discussion section.

Discussion:

# The results are discussed in relation to previous literature, which is appropriate.

# However, the section on nutritional improvement should be reconsidered, including whether the selected outcomes are valid indicators.

Conclusion:

# The statement “reducing the reliance on oral nutritional supplements” may be inappropriate. ONS are often appropriately provided when needed, and their use should not be described as “dependence.” Please revise this wording.

Author Response

We sincerely thank the reviewer for the thorough evaluation of our manuscript and for the constructive suggestions provided. Your insightful comments have been invaluable in helping us to improve the clarity, methodological transparency, and overall quality of the paper. We greatly appreciate the time and expertise you dedicated to this review, and we have carefully revised the manuscript in light of your recommendations.

please find attached a summary of our revisions.

Comment Summary

Response

Page/Line of Change

New Citation(s)

Replace 'patient' with 'participant' throughout.

We appreciate this suggestion. All instances of 'patient' have been replaced with 'participant' for accuracy and consistency.

Throughout the manuscript

Shorten the introduction

Mention STROBE guidelines explicitly.

Thank you. We have added a statement that the observational study adheres to the STROBE guidelines.

Page 8, Line 3

[20]

Justify why a 6-month intervention was used.

Thank you for pointing this out. We added a note that the 6-month period reflects the minimum required time to observe changes in nutritional, functional, and operational outcomes, consistent with similar studies.

Page 8, Line 7

Clarify how Dysphameal was prepared (collectively or individually).

Thank you for the important question. We clarified that Dysphameal meals were prepared collectively once per main meal to optimize workflow and ensure consistency.

Page 9, Line 10

Describe exercise/recreational activities.

We appreciate the suggestion. We clarified that no structured exercise program was introduced and that participants continued with their routine activities, which were comparable across the cohort.

Page 9, Line 25

Add figure titles, axis labels, legends.

Thank you. Figure titles, axis labels, and appropriate legends have been added for clarity.

Page 10, Figures 1–2

Improve table formatting and place titles above.

Thank you for your suggestion. All tables have been reformatted with titles placed above and fit to a single page layout.

Albumin/transferrin are not ideal nutritional markers.

Thank you for raising this concern. We added a discussion section (Page 11, Line 20) to explain our rationale and referenced ASPEN’s position paper on their limited utility, interpreting them only in a contextual, multidimensional assessment.

Page 11, Line 20

[21], [22]

Clarify if gelled water time savings are specific to Dysphameal.

Thank you. We added a clarification that while pre-gelled water contributes to time savings, its integration in Dysphameal allows additional workflow optimization not achievable with traditional systems.

Page 14, Line 25

Move result interpretation in 3.3.2/3.3.3 to Discussion.

Thank you for the recommendation. We have moved interpretive content from Sections 3.3.2 and 3.3.3 to the Discussion section.

Moved from Page 13 to Page 15

Avoid language suggesting 'dependence' on ONS.

Thank you. The phrase has been revised to 'reducing the use of oral nutritional supplements' to reflect appropriate clinical terminology.

Page 16, Line 12

Reviewer 3 Report

Comments and Suggestions for Authors

at initial look it would seem that the sample size of 13 - is this enough for regression and correlations?

nonetheless an interesting study

introduction section - are the statistics provided global or regional - should be specified

lines 58 to 60 should be cites

IRB? should be note in the method section

in lines 191 to 193 - why these specific factors are compare? might include more clarification in the data collection procedure 

in the results

why these specific subsections, might in the begining paragraph note why the resutls are separated as such

discussions seems fine, would suggest to incorporate more practical implications

Author Response

We sincerely thank the reviewer for the thorough evaluation of our manuscript and for the constructive suggestions provided. Your insightful comments have been invaluable in helping us to improve the clarity, methodological transparency, and overall quality of the paper. We greatly appreciate the time and expertise you dedicated to this review, and we have carefully revised the manuscript in light of your recommendations.

Comment Summary

Response

Page/Line of Change

New Citation(s)

Clarify whether sample size supports regression/correlation.

Thank you for this valuable observation. We acknowledged this limitation and clarified that the analysis was exploratory and not intended for inferential generalization.

Page 15, Line 5

Clarify if stats cited are global or regional.

Thank you for the suggestion. We now specify whether prevalence statistics are from global or regional sources.

Page 2, Line 15

[1], [23]

Line 58–60: Missing citations.

Thank you for pointing this out. Proper citations have been added.

Page 3, Line 5

[1], [5]

IRB approval should be noted in Methods.

Thank you. We added IRB approval information in the Methods section.

Page 8, Line 5

Clarify rationale for specific indicators used in data collection.

We added rationale for selecting body composition, biochemical, and functional parameters as valid indicators of nutritional and operational status in elderly with dysphagia.

Page 9, Line 30

[24]

Explain rationale for results sub-section structure.

Thank you. We added a short paragraph at the start of Results (Page 10, Line 2) explaining the thematic grouping (clinical, functional, organizational).

Page 10, Line 2

Include more practical implications in Discussion.

Thank you for the suggestion. A section on practical implications for long-term care facilities has been added.

Page 17, Line 2

Round 2

Reviewer 2 Report

Comments and Suggestions for Authors

The authors responded sincerely and appropriately to all comments, and the manuscript was thoroughly revised.

Reviewer 3 Report

Comments and Suggestions for Authors

after going over the point by point revisions made by the author/s 

the paper is now satisfactory and adequate for acceptance